# Adipose Tissue Stem Cells (ASCs) and ASC-Derived Extracellular Vesicles Prevent the Development of Experimental Peritoneal Fibrosis

**DOI:** 10.3390/cells14060436

**Published:** 2025-03-14

**Authors:** Priscila Q. Gouveia, Camilla Fanelli, Felipe M. Ornellas, Margoth R. Garnica, Ana L. R. Francini, Gilson M. Murata, Luiz H. G. Matheus, Marcelo M. Morales, Irene L. Noronha

**Affiliations:** 1Laboratory of Cellular, Genetic, and Molecular Nephrology, Renal Division, Medical School, University of São Paulo, São Paulo 01246-903, Brazil; pri_queiroz_gouveia@outlook.com (P.Q.G.); camilla.fanelli@usp.br (C.F.); fmornellas@gmail.com (F.M.O.); margoth.ramos@hc.fm.usp.br (M.R.G.); gilson.murata@usp.br (G.M.M.); 2Laboratory of Carbohydrate and Radioimmunoassay, School of Medicine, University of São Paulo, São Paulo 01246-903, Brazil; luiz.h.g.matheus@gmail.com; 3Laboratory of Cellular and Molecular Physiology, Carlos Chagas Filho Institute of Biophysics, Federal University of Rio de Janeiro, Rio de Janeiro 21941-853, Brazil; mmorales@biof.ufrj.br

**Keywords:** mesenchymal stromal cells, adipose-tissue-derived stem cells (ASCs), extracellular vesicles (EVs), peritoneal fibrosis

## Abstract

Cell therapy utilizing mesenchymal stromal cells (MSCs) through paracrine mechanisms holds promise for regenerative purposes. Peritoneal fibrosis (PF) is a significant complication of peritoneal dialysis. Various strategies have been proposed to protect the peritoneal membrane (PM). This study explores the effectiveness of adipose-tissue-derived stem cells (ASCs) and extracellular vesicles (EVs) at mitigating PF using a rat model of PF induced by chlorhexidine gluconate. ASC and EV treatments effectively prevented an increase in the thickness of the PM and diminished the number of myofibroblasts, fibronectin expression, collagen III expression, and PF-related factors such as TGF-β and FSP-1. Smad3 gene expression decreased in the treatment groups, whereas Smad7 gene expression increased in treated animals. In addition, ASC and EV injections showed potent anti-inflammatory effects. Glucose transport through the PM remained unaffected in relation to the PF group; both treatments promoted an increase in ultrafiltration (UF) capacity. The PF+EVs treated group showed the highest increase in UF capacity. Another critical aspect of ASC and EV treatments was their impact on neoangiogenesis in the PM which is vital for UF capacity. Although the treated groups displayed a significant decrease in VEGF expression in the PM, peritoneal function remained effective. In conclusion, within the experimental PF model, both ASC and EV treatments demonstrated anti-inflammatory effects and comparably hindered the progression of PF. The EV treatment exhibited superior preservation of peritoneal function, along with enhanced UF capacity. These findings suggest the potential of ASCs and EVs as novel therapeutic approaches to prevent the development of PF associated with peritoneal dialysis.

## 1. Introduction

Peritoneal dialysis (PD) serves as a high-quality renal replacement therapy in the treatment of stage 5 chronic kidney disease (CKD). Recent estimates suggest that more than 390,000 patients receive PD worldwide, representing approximately 11% of the global dialysis population [1]. However, long-term use of PD introduces a complex interplay of challenges resulting from structural and functional adaptations within the peritoneal membrane (PM), ultimately compromising treatment efficacy [2,3]. These adaptations manifest as peritoneal inflammation, neoangiogenesis, and peritoneal fibrosis (PF), collectively impairing ultrafiltration capacity. Effective prevention or reversal of adverse events associated with extended PD remains elusive, leading to therapeutic failure [2].

Efforts to mitigate PF predominantly revolve around biocompatible dialysis solutions, with limited success [4]. Recent investigations have highlighted the potential of tamoxifen and rBMP7 of attenuating PF within experimental uremic rat models [5]. These findings suggest that interventions targeting the transforming growth factor-β (TGF-β) signaling pathway and enhancing Smad7 expression could offer strategies against the development of PF. The intricate TGF-β pathway, orchestrated by Smad co-factors, involves the migration of the Smad2/3/4 complex to the cell nucleus, promoting transcription of pro-fibrotic genes. Conversely, Smad7 counters these effects, inhibiting TGF-β/Smad signaling. Hence, modulating Smad expression has emerged as a potential therapeutic avenue for PF [6,7,8].

Stem-cell-based therapies also offer an alternative approach to quell the inflammatory and fibrotic processes, potentially inducing the regeneration of the PM [9,10,11]. Although previous studies have demonstrated the feasibility and therapeutic effects of adipose-tissue-derived stem cells (ASCs) in experimental models [9,10,11] and clinical applications targeting PF [12], the potential of extracellular vesicles (EVs) derived from ASCs compared with the cells themselves remains insufficiently explored. EVs, as a cell-free therapeutic strategy, retain the paracrine regenerative effects of ASCs and offer advantages such as improved scalability for therapeutic production and enhanced stability for storage and transport, all of which are critical for future clinical applications.

A robust body of evidence highlights the paracrine attributes of mesenchymal stromal cells (MSCs) and their secretome as potent mediators of regenerative and immunomodulatory effects [13,14,15], rather than their transdifferentiation regenerative effects [16,17]. MSC-conditioned media contain a variety of bioactive soluble factors shed from the source cells into the extracellular space, collectively termed EVs, involved in an intricate network.

EVs play a pivotal role in facilitating cell-to-cell communication, delivering essential content such as RNAs, miRNAs, DNAs, lipids, and proteins to target cells in a paracrine manner [15,18,19]. Through this mode of communication, EVs influence gene expression, cell proliferation, immunomodulation, angiogenesis, and anti-apoptotic responses in target cells [20]. Notably, EVs serve as vital mediators of the renoprotective effects observed in preclinical studies of kidney diseases, reflecting improvements in renal parameters [21,22,23]. In the clinical arena, a pilot study involving patients with stage 3 and 4 CKD has demonstrated the safety and potential benefits of EVs derived from umbilical cord MSCs [24].

By navigating these multifaceted challenges, our investigation aims to unveil the prophylactic effects of ASCs and ASC-derived EVs in an experimental rat model of PF induced by chlorhexidine gluconate (CG) administration. This model replicates key aspects of PF progression, encompassing PF and inflammatory infiltration [5,25,26].

## 2. Materials and Methods

### 2.1. Animals

Thirty-five male Wistar rats (weight 280–320 g, age 10–12 weeks) were obtained from an established colony at the University of São Paulo, Brazil. They were kept under a constant temperature (23 °C ± 2 °C) and under a 12 h light/dark cycle, and had free access to conventional rodent chow and tap water. Thirty-three Wistar rats were used in the experimental protocol and two were used as adipocyte tissue donors for isolation of MSCs.

All animal protocol studies were approved by the Ethics Committee for Analysis of Research Projects (FMUSP-CAPPesq 165/15) and were performed in accordance with our institutional guidelines and with international regulations for the manipulation and care of experimental animals.

### 2.2. Adipose-Tissue-Derived Mesenchymal Stem Cells

Adipose tissue from epididymal fat pads from two adult male Wistar rats was obtained as described previously [11,27]. The adipose tissue samples were minced with sterile scissors and digested with 0.075% collagenase type IA (Sigma-Aldrich, St Louis, MO, USA) at 37 °C for 40 min, under constant stirring. Subsequently, the collagenase was inactivated with fetal bovine serum (FBS; Gibco by Life Technologies, Carlsbad, CA, USA) and the cell suspension was washed with a phosphate buffer (PBS) and centrifuged twice, for 5 min, at 260× *g*. After centrifugation, the isolated cells were cultured (37 °C, 5% CO_2_, 20% O_2_, and 95% humidity) in T75 culture flasks (Greiner Bio-one, Essen, Germany) with Dulbecco’s modified Eagle’s medium (DMEM; Gibco by Life Technologies) containing 15 mM HEPES (Gibco by Life Technologies), 10% inactivated FBS, 100 units/mL penicillin, and 100 mg/mL streptomycin antibiotic solution (Gibco by Life Technologies). The culture medium was changed three times a week, and nonadherent cells were removed. Adherent cells reaching 80% confluence were passaged with a 0.05% Trypsin–EDTA solution (Gibco by Life Technologies) at 37 °C and then maintained in DMEM with 10% FBS, 100 units/mL penicillin, and 100 mg/mL streptomycin antibiotic solution (complete medium). Then, 2 × 10^6^ fresh ASCs were diluted in PBS and used for characterization. Animals in the PF/ASC group received two doses of about 2 × 10^6^ fresh ASCs administered intraperitoneally on the 3rd and 10th days after PF induction.

At the fourth passage, ASCs were characterized by: (1) adherence to plastic (Appendix A), (2) being negative for the hematopoietic cell surface marker CD45 and positive for markers CD29, CD90, CD44, and CD105 (eBioscience, San Diego, CA, USA), detected by flow cytometry using FacsCantoII (Becton Dickinson, Franklin Lakes, NJ, USA) (Appendix A), and (3) the ability to differentiate into adipocytes, chondrocytes, and osteoblast-like cells [28]. Differentiation of ASCs was carried out using a STEMPRO differentiation kit (Life Technologies, New York, NY, USA). After 21 days, oil red O was used for fatty vacuoles to assess adipogenic differentiation, alizarin red staining was used for calcium phosphate deposits to evaluate osteogenic differentiation, and alcian blue was used for sulfated proteoglycans to confirm chondrogenic differentiation (Appendix A).

### 2.3. Isolation of EVs

EVs were obtained from the supernatants of ASCs cultured at the fourth passage in DMEM after 24 h of FBS deprivation, as previously described [29,30]. ASC viability was established by trypan blue exclusion (>97%) and by the absence of apoptotic cells detected by propidium iodate assay (eBioscience) (Appendix A). Twenty milliliters of the conditioned medium of a confluent culture flask containing 2 × 10^6^ ASCs was removed, and EVs were enriched as follows: centrifugation at 300× *g* for 30 min to remove cells and debris in the suspension. Subsequently, the supernatant was ultracentrifuged at 100,000× *g* for 3 h (Himac CP80NX ultracentrifuge and fixed-angle rotor; P50AT2 Hitachi Koki Co., Tokyo, Japan). The EVs containing sediment were washed with PBS and subjected to a second ultracentrifugation at 100,000× *g* for 20 min. Isolated EVs were resuspended in 2 mL of PBS and used for characterization and inoculation into the animals.

### 2.4. Characterization of EVs

The size and morphological characteristics of the EVs derived from ASCs were analyzed by transmission electron microscopy. EVs were fixed in 2.5% glutaraldehyde in 0.1 M sodium cacodylate buffer (pH 7.4) for 40 min and post-fixed with a solution of 1% OsO_4_, 0.8% potassium ferricyanide, and 2.5 mM CaCl_2_ in the same buffer for 20 min at room temperature, then dehydrated in an ascending series of acetone and embedded in PolyBed 812 resin. Ultrathin sections were stained with uranyl acetate and lead citrate; these sections were examined under a JEM 1011 JEOL transmission electron microscope, at 80 kV (Carl Zeiss NTS, Oberkochen, Germany).

The size distribution and concentration of EVs in the solution were determined using a NANOSIGHT 3 nanoparticle tracking analysis (NTA) device (NanoSight, Malvern, UK), equipped with an SCMOS camera and NTA software (version NTA 2.0 Build). For each sample, six 30 s videos were recorded and analyzed. The dilution factor was applied to calculate the total particle concentration. Additionally, EVs were characterized by dot blot analysis for the tetraspanins CD9 and CD63. For this purpose, 10 µL of EV solution, containing approximately 10 µg of total protein, was spotted onto nitrocellulose membranes and incubated with primary antibodies (rabbit anti-CD9, Sigma-Aldrich #C9993, and rabbit anti-CD63, Abcam #Ab134045). Membranes were subsequently treated with anti-rabbit HRP-conjugated secondary antibodies and developed using an ECL chemiluminescent substrate.

### 2.5. Experimental Design

PF was induced in rats by daily intraperitoneal injections of CG at 0.1% in 15% ethanol dissolved in saline for 30 alternate days, as described previously [5]. Tramadol hydrochloride (5 mg/kg) was administered intramuscularly before CG injections to prevent any discomfort.

Animals were divided into four groups: control (*n* = 8), receiving only saline solution intraperitoneally, PF (*n* = 9), receiving CG to induce PF, PF+ASCs (*n* = 8), PF rats receiving 2 × 10^6^ ASCs administered through intraperitoneal injections on the 3rd and 10th days after PF induction, and PF+EVs (*n* = 8), PF rats receiving approximately 4 × 10^11^ EVs obtained from the culture supernatant of 2 × 10^6^ ASCs administered through intraperitoneal injections on the 3rd and 10th days after PF induction.

All animals from the control, PF, PF+ASCs, and PF+EVs groups were followed for 30 days after the beginning of the experiments. After 30 days, the peritoneal function was analyzed, and the animals were euthanized under deep anesthesia and sedation by intraperitoneal injection of a ketamine hydrochloride solution (70 mg/kg) (Ketamin-S; Cristália, São Paulo, Brazil) and xylazine (11 mg/kg) (Ronpun; Bayer, Leverkusen, Germany). PM samples collected from the left anterior abdominal wall were carefully dissected, frozen immediately in liquid nitrogen, and stored at −80 °C for polymerase chain reaction (PCR) assays. Additional sections for histologic analysis were fixed in a Duboscq-Brazil solution (see Section 2.7 and Section 2.8).

### 2.6. GC Toxicity on ASCs

In vitro experiments were carried out to analyze whether CG could exert a cytotoxic effect on the ASCs by incubating ASCs with CG at a concentration of 0.1% added to the cell culture medium. After 2 h of incubation, apoptosis was evaluated using a propidium iodide assay (Molecular Probes, Eugene, OR, USA), following the manufacturer’s manual. Subsequently, a flow cytometric analysis was performed using a BD FACSCanto II (Becton Dickinson, Franklin Lakes, NJ, USA) and the BD FACSDiva™ software version 9.0 (Becton Dickinson, Franklin Lakes, NJ, USA) (Appendix A).

### 2.7. PM Histomorphometry

PF was evaluated from 2–3 µm thick sections of PM stained with the Masson’s trichrome technique. At least ten pictures at a 200× final magnification were taken from each rat, and the thickness of the PM in all photomicrographs was measured using Image-Pro Plus Software 7.0 (Media Cybernetics, Bethesda, MD, USA). The mean peritoneal thickness from each rat was then calculated.

### 2.8. Immunohistochemistry and Immunofluorescence

Further 4 µm thick PM sections were deparaffinized and subjected to antigen retrieval by microwave irradiation in a citrate buffer (pH 6.0). The streptavidin–biotin–alkaline phosphatase technique was used for the immunohistochemistry, as described previously [5,31]. The endogenous biotin activity was blocked, and slides underwent non-specific blocking with non-immune rabbit and/or goat IgG solution. Samples were incubated overnight with the following primary antibodies: monoclonal mouse anti-rat CD68 clone ED-1 (Serotec, Raleigh, NC, USA) for macrophages, monoclonal mouse anti-rat CD43 (Serotec), considered to be a T-cell-associated marker, and monoclonal mouse anti-α smooth muscle actin (α-SMA) (Sigma, St Louis, MO, USA) for myofibroblasts. Secondary biotinylated anti-mouse IgG (Vector Laboratories, Newark, CA, USA) and the streptavidin–biotin–alkaline phosphatase complex were used subsequently. Slides were developed with the fast red solution (Sigma) and counterstained with Mayer’s hematoxylin (Merck, Darmstadt, Germany). A quantitative analysis of immunohistochemistry was performed in a blinded fashion, under 200× microscopic magnification. Peritoneal macrophages and the inflammatory infiltrate population were counted in at least 20 microscopic fields/animal and expressed as cells/mm^2^. The percentage area of the PM occupied by α-SMA was calculated by measuring both the total PM area and the positive PM area (stained in red after immunohistochemistry development) of at least 20 microscopic fields from each studied animal using the Image-Pro Plus 7.0 software (Media Cybernetics, Rockville, MD, USA). These two measurements were used to calculate the % α-SMA for each microscopic field, and the arithmetic mean of all fields from the same animal was used to obtain the individual % α-SMA.

Indirect immunofluorescence was carried out on paraffin PM sections to analyze the M1 and M2 macrophage subpopulations. After deparaffinization, renal samples were subjected to antigen retrieval by microwave irradiation with Tris EDTA buffer (pH 9.0) + 0.05% Tween. Non-specific protein binding was blocked with a 10% bovine albumin fraction for 30 min at room temperature. The sections were then incubated with the following primary antibodies: anti-CD68 diluted at 1:50 and anti-mannose receptor antibody (CD206; Abcam, San Francisco, CA, USA) diluted at 1:800 overnight at 4 °C. After washing, the slides were incubated with Alexa-Fluor-labeled secondary antibodies (Alexa 546 anti-rabbit and Alexa 488 anti-mouse; Life Technologies, Eugene, OR, USA) for 1 h at room temperature. The slides were ten incubated for 5 min with DAPI diluted at 1:1000 (Life Technologies, Carlsbad, CA, USA) for nuclear staining. After washing, the slides were mounted with glycerol 1:1 in PBS, and the sections were analyzed under an epifluorescence microscope (Nikon Eclipse 80i; Nikon Instruments, Tokyo, Japan) at 400× magnification, using a 488 nm filter for CD68 (green), 546 nm filter for CD206 (red), and 504 nm filter for DAPI (blue). To evaluate the presence of the M1 and M2 macrophage subtypes in the PM samples from the experimental groups, 20 epifluorescence microscopic fields were analyzed per animal, under 200× magnification, using the NIS-Elements software version 6.10.01 (Nikon, Tokyo, Japan). Each microscopic field was captured under three different light filters (blue, green, and red), thus producing three microphotographs for each field. Using the “merge” tool of the NIS-Elements software version 6.10.01 (Nikon, Tokyo, Japan), a fourth picture per field was produced with an overlay of all colors. The M1 macrophage subpopulation was recognized as CD68+/CD206− cells, visible in the merged pictures as green cells. The number of green cells was counted and expressed as M1 cells/mm^2^. M2 macrophages were recognized as CD68+/CD206+ cells, visible in the merged pictures as colorful yellowish cells due to the overlay of green, red, and blue from the nuclei in the same cell. The number of colorful cells was counted and expressed as M2 cells/mm^2^. To express the percentage of M1 and M2 macrophages in PM samples, we used the following calculations: (M1 × 100)/total macrophages and (M2 × 100)/total macrophages, respectively.

### 2.9. Quantitative Assessment of Angiogenesis

The endothelial marker DyLight 594-labeled *Griffonia simplicifolia* isolectin B4 (Vector Laboratories) was used to detect freshly formed blood vessels and analyze neoangiogenesis by immunofluorescence [5]. Nuclei were counterstained with DAPI (Life Technologies). Ten microscopic fields were scored under 400× magnification using a 594 nm filter for isolectin and a 540 nm filter for DAPI (Nikon Eclipse 80i microscope). The number of isolectin-B4-positive blood vessels was counted, and the density of capillaries present in each slide was determined by dividing the number of blood vessels by the area of the PM and expressed as the number of vessels/mm^2^.

### 2.10. Real-Time Quantitative PCR

A quantitative real-time reverse transcription PCR was performed to measure the relative gene expression of TGF-β, an S100 calcium-binding protein A4 gene that codifies fibroblast-specific protein-1 (FSP-1), Smad3, Smad7, TNF-α, IL-1β, IL-6, fibronectin, collagen III, and VEGF, in peritoneal samples of the studied animals as described previously [32]. The PM from the anterior abdominal area was collected in cryotubes, flash-frozen by immersion in liquid nitrogen, and stored at −80 °C. The total RNA was extracted using Trizol (Ambion by Merck). Constitutive gDNA was eliminated from the RNA samples using a turbo dnase-free kit (Invitrogen, Waltham, MA, USA). The RNA concentration was measured by spectrophotometry in a Nanodrop ND-2000. First-strand cDNA was synthesized from the total RNA using the M-MLV RT enzyme (Promega, Madison WI, USA) for cDNA. Relative mRNA levels were measured with the Syber Green qPCR Super Mix Universal (Invitrogen) and StepOne Plus equipment (Applied Biosystems, Waltham, MA, USA). All experiments were performed in triplicates. Real-time PCR conditions were as follows: 50 °C for 5 min, 95 °C for 5 min, 95 °C for 15 s, 60 °C for 30 s, and 72 °C for 30 s, with analysis of the fluorescence emission at 60 °C. Forty cycles were performed for each experiment. Quantitative comparisons were obtained using the ΔΔCT method (Applied Biosystems). The 18S was used as a housekeeping endogenous control. Primer sequences for the amplification of the target genes are summarized in Appendix A.

### 2.11. Peritoneal Function

On day 30, before euthanasia, 0.09 mL/g of a 4.25% peritoneal dialysis solution (Baxter Solution, São Paulo, Brazil) was administered to the animals intraperitoneally. Two hours later, the rats were anesthetized and the peritoneal fluid was collected with a disposable sterile syringe under conditions for ultrafiltration (UF) measurements. UF values were determined by subtracting the drained volume from the infused volume. Peritoneal fluid samples were centrifuged at 1500 rpm for 5 min, after which the glucose levels were measured (Cobas C111 Analyzer; Roche, Indianapolis, IN, USA). The mass transfer of glucose from the peritoneum was determined using the following formula: (initial dialysate glucose × initial volume) − (final glucose × final volume) [33].

### 2.12. Statistical Analysis

Data are presented as means ± standard error (SEM), and statistical analyses were performed using GraphPad Prism 6 for Windows (GraphPad Software, La Jolla, CA, USA). One-way ANOVA with Tukey’s post-hoc test was used to compare the groups. *p* values < 0.05 were considered to be statistically significant.

## 3. Results

### 3.1. Isolation and Characterization of ASCs and EVs

ASCs were isolated and cultured as per the established protocols in our laboratory, respecting previously established criteria. ASCs in the fourth passage underwent immunophenotyping by flow cytometry and cell plasticity tests, as shown in the Appendix A. Transmission electron microscopy revealed that the obtained EVs predominantly exhibited spherical structures with varying diameters (Appendix A). The nanoparticle tracking analysis confirmed that each EV dose comprised approximately 4 × 10¹¹ particles, with sizes ranging from 50 nm to 600 nm, and a predominant population of particles with an approximate diameter of 250 nm (mode) (Appendix A). The dot blot analysis demonstrated that EV samples were positive for the tetraspanins CD9 and CD63 (Appendix A). 

### 3.2. ASC Viability

The rationale for conducting this analysis is explained in Section 2. The viability of the ASCs was 97% ± 0.5% after culture for 24 h without FBS. Apoptosis of ASCs was around 35% ± 0.7% in the presence of 0.1% CG in the culture medium after 24 h (Appendix A).

### 3.3. PF Experimental Model

The experimental model of PF was successfully established with local injections of CG in adult Wistar rats. A histological analysis of the PM samples from PF animals revealed a marked increase in the thickness of the PM, particularly in the submesothelial compact zone. This increase was accompanied by extracellular matrix deposition, heightened cell proliferation, and pronounced inflammatory cell infiltration, in contrast to control animals which displayed a preserved PM structure (Figure 1 and Appendix A).

### 3.4. ASC and EV Treatments Were Equally Efficient in Protecting the Development of FP

We compared the therapeutic effects of ASCs on PF with those achieved through the administration of ASC-derived EVs. Intraperitoneal inoculations of 2 × 10^6^ ASCs or 30 µg of EVs (~4 × 10^11^ particles) were administered to rats in the CG-induced PF model. The treatments were given in two doses at the same time intervals (3rd and 10th days after the beginning of PF induction). Treatment with ASCs or ASC-EVs significantly prevented the thickening of the PM in the PF+ASCs and PF+EVs groups, respectively (Figure 1 and Appendix A).

### 3.5. Administration of ASC or EV Decreased the Number of Myofibroblasts and Reduced the Expression of Factors Involved in PF

Myofibroblasts, key cells in fibrogenesis, were significantly more numerous in the PF group compared with the control group. PF animals treated with either ASCs or EVs showed a striking reduction in this biomarker, reaching levels similar to those observed in the control group (Figure 2 and Appendix A). The impact of ASCs and EVs on reducing extracellular matrix components in the PF model was demonstrated by PCR analysis of fibronectin and collagen III, in parallel with the histological findings. PF animals demonstrated a significant overexpression of both fibrosis-associated genes, a phenomenon counteracted by treatments with ASCs or EVs. Although ASCs induced a noteworthy reduction in fibronectin and collagen III peritoneal expression, EVs fully normalized the expression of both genes (Figure 3A,B and Appendix A).

### 3.6. Anti-Fibrotic Effects of ASCs and EVs Are Mediated by the TGF-β/Smad Pathway

The main mechanisms possibly involved in fibrogenesis were analyzed. The quantitative PCR analysis revealed a significant gene overexpression of TGF-β, a major fibrosis mediator, and FSP-1, a fibroblast biomarker, in the PM of untreated PF animals. The local expression of both genes was normalized in a comparable manner by ASC and EV treatments in the PF+ASCs and PF+EVs groups, respectively (Figure 3C,D and Appendix A). To further understand the protective mechanisms of ASCs and EVs against PF, TGF-β signaling-related genes, Smad3 and Smad7, were analyzed. Animals subjected to the PF model exhibited local Smad3 gene overexpression, but this was not observed in PF rats treated with either ASCs or EVs. Expression of the inhibitory factor, Smad7, did not differ between the control, on one hand, and the PF groups, on the other, but showed a significant increase in PF+ASCs and PF+EVs animals (Figure 3E,F and Appendix A).

### 3.7. Anti-Inflammatory Effects of ASCs and EVs

Characterization of the cellular inflammatory infiltrate in the PM samples showed the presence of macrophages (CD68+ cells) and leukocyte infiltration (CD43+ cells). Untreated PF animals displayed significant PM inflammation, demonstrated by marked macrophage and leukocyte infiltration. Treatment with ASCs and EVs significantly attenuated PM leukocyte infiltration, with a similar effect observed among the PF+ASCs and PF+EVs groups, indistinguishable from the control group (Figure 4 and Appendix A). 

Despite reducing the global macrophage infiltration (represented by the sum of M1 and M2 macrophage subtypes) in the PM samples of PF rats (Figure 5Q), only the reduction in the M1 subpopulation was statistically and significantly different in the PF+ASCs and PF+EVs groups compared with the untreated PF group, suggesting a relative conservation of M2 macrophages in the treated animals. The bar graphs in Figure 5R show that the percentage of M2 macrophages in untreated PF animals (36.1%) seems to have been increased by the treatments with ASCs and EVs (46.3% and 45.7%, respectively).

Moreover, as can be seen in Figure 6, PF animals exhibited a local mRNA overexpression of IL-1β, TNF-α and IL-6, which were normalized with both ASC or EV treatments.

### 3.8. Effects of ASCs and EVs on PM Angiogenesis

The occurrence of neoangiogenesis in the PM of all groups was assessed by determining capillary vessel density in paraffin-embedded PM sections using the endothelial marker DyLight-594-labeled isolectin B4, which detects newly formed blood vessels. A pronounced increase in PM capillary density was observed in untreated PF animals, whereas treatments with ASCs or EVs reduced and normalized the capillary density in the PM of rats that underwent experimental PF (Figure 7 and Appendix A). Similarly, VEGF gene expression, a vascular and endothelial growth factor, was found to have increased in untreated PF animals, whereas the ASC and EV treatments significantly decreased VEGF gene expression (Figure 7 and Appendix A).

### 3.9. ASCs and EVs Preserved PM Function

To determine whether the morphological preservation of the PM as a result of treatment with ASCs and EVs had an impact on the preservation of peritoneal function in uremic rats with PF, we analyzed the UF rate and the mass transfer of glucose (MTG). Untreated PF animals showed a lower UF rate and higher MTG values compared with the control group. There was no significant improvement in these parameters following ASC treatment in both analyses. However, administration of EVs to animals subjected to PF promoted an improvement in the UF rate (Figure 8A,B and Appendix A).

## 4. Discussion

In this study, we investigated the protective effects of ASCs and EVs against PF induced by CG. We demonstrate that both treatments were effective at reducing histological, molecular, and functional parameters associated with PF. EVs showed more pronounced results than ASCs, highlighting their differentiated therapeutic potential.

Our PF model resulted in significant thickening of the PM and increased deposition of extracellular matrix, particularly fibronectin and collagen III, in the submesothelial layer, confirming the fibrotic phenotype described previously [26]. Both ASCs and EVs reduced these markers, but EVs were more effective, particularly at normalizing the expression of fibronectin and collagen III. These findings corroborate that EVs can modulate fibrogenesis, as reported in other studies [23,24,27,34]. This is a significant distinction, as previous studies do not directly evaluate the impact of adipose-tissue-derived EVs in this experimental PF model, making our study unique in this regard.

The TGF-β/Smad signaling pathway was strongly modulated by both treatments. We observed that both ASCs and EVs reduced the expression of TGF-β and Smad3, two central mediators of PF, and increased the expression of Smad7, a natural inhibitor of TGF-β. These findings are consistent with the literature, which describes the role of Smad7 overexpression in inhibiting fibrogenesis [7,8,34,35,36,37,38,39,40,41,42]. The results suggest that EVs, by modulating this pathway effectively, play an important role in interrupting pro-fibrotic molecular mechanisms, highlighting their relevance as therapeutic mediators.

In addition to the effects on fibrogenesis, our data demonstrate significant immunomodulatory properties. We observed a reduction in the infiltration of M1 (pro-inflammatory) macrophages and leukocytes in both treatment groups, but EVs also preserved an anti-inflammatory environment, demonstrated by the maintenance of M2 macrophages, which have reparative functions. This modulation of macrophage phenotype could be a key mechanism in the protection against fibrosis, as suggested by studies investigating the plasticity of these cells in chronic inflammatory microenvironments [28,29,32,35,43,44]. The specific role of EVs in altering this balance needs to be further explored, especially in the context of fibrotic diseases.

Neoangiogenesis, often associated with peritoneal dysfunction, was significantly reduced by the treatments, with a more robust effect observed in the EV group. Treated groups showed reduced capillary density and VEGF expression, a key factor in pathologic angiogenesis [2]. This finding is important because neoangiogenesis is implicated in UF dysfunction in chronic PF models. In addition, although ASCs did not have a significant impact on functional preservation, EVs were able to improve UF, suggesting a direct functional benefit from this treatment.

A key point that stands out is the differential effect of EVs compared with ASCs in the therapeutic context. EVs not only replicate many of the beneficial effects of ASCs, but also offer practical advantages, particularly considering future clinical applications, such as greater stability, scalability for production, and lower risk of associated complications. These aspects, together with the results of this study, place EVs in a promising position as a therapeutic alternative for PF and, potentially, other fibrotic conditions [45].

However, we acknowledge the limitations of our study. We did not evaluate intermediate time points, which could provide a more detailed view of the dynamic mechanisms of treatment. Furthermore, the complete molecular profile of EVs, including the role of microRNAs and specific proteins, still needs to be investigated. These molecules could provide valuable insights into the precise mechanisms by which adipose-tissue-derived EVs modulate fibrogenesis, inflammation, and angiogenesis, opening new opportunities for therapeutic interventions.

## 5. Conclusions

In summary, our study demonstrates that adipose-tissue-derived EVs represent a highly promising therapeutic alternative for preventing PF. The observed effects include significant reductions in fibrogenesis, inflammation, and neoangiogenesis, with a direct functional impact on preserving UF. These results highlight the role of EVs as innovative therapeutic mediators and provide a solid foundation for future preclinical and translational investigations.

## Figures and Tables

**Figure 1 cells-14-00436-f001:**
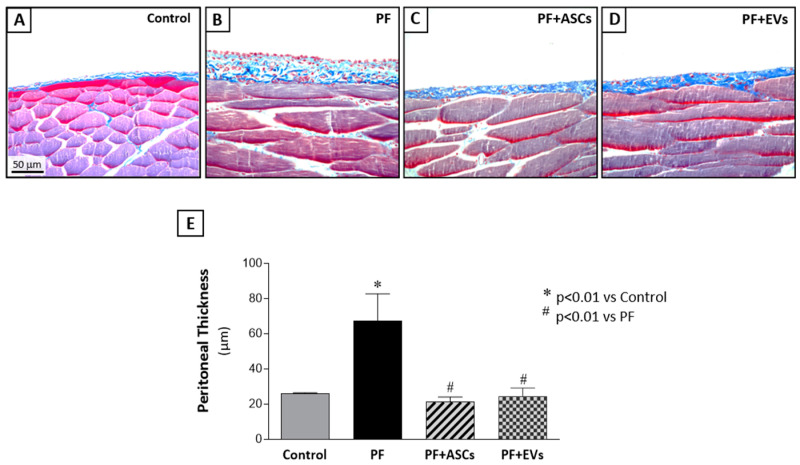
Histological features of peritoneal biopsy specimens obtained from the animals in the different groups, stained with Masson’s trichrome (×400). (**A**) The control group (*n* = 6) presented normal peritoneal morphology with no morphological alterations in mesothelial, submesothelial, or muscle tissue. (**B**) The PF group (*n* = 8) presented a marked thickening of the peritoneal membrane (PM), accompanied by increased cellularity and collagen deposition (stained in blue). Treatments with (**C**) ASC (*n* = 7) or (**D**) EVs (*n* = 8) reduced PM thickening and fibrosis, preserving the peritoneal morphology in the PF+ASCs and PF+EVs groups. (**E**) Bar graph of the quantification of PM thickening in all experimental groups on day 30 after PF induction. A one-way ANOVA statistical analysis was performed: * *p* < 0.01 vs. control, # *p* < 0.01 vs. PF.

**Figure 2 cells-14-00436-f002:**
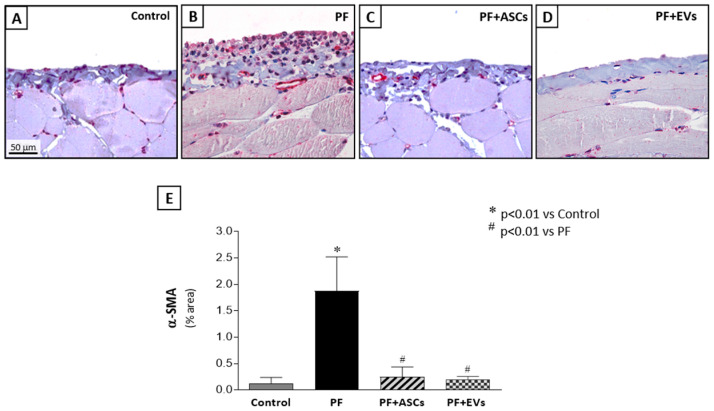
Myofibroblasts were detected in the peritoneal membrane samples of all groups through the positive expression of α-smooth muscle actin (α-SMA), using immunohistochemistry (×400). (**A**) Control rats (*n* = 6) showed few α-SMA positive cells. (**B**) PF (*n* = 8) induced by chlorhexidine gluconate was associated with a marked accumulation of myofibroblasts in the PF group that was drastically reduced by treatments with (**C**) PF+ASCs (*n* = 7) or (**D**) PF+EVs (*n* = 8). (**E**) Results are presented as a bar graph of the results of a quantitative analysis of the α-SMA-positive peritoneal area in the experimental groups. A one-way ANOVA statistical analysis was performed: * *p* < 0.01 vs. control, # *p* < 0.01 vs. PF.

**Figure 3 cells-14-00436-f003:**
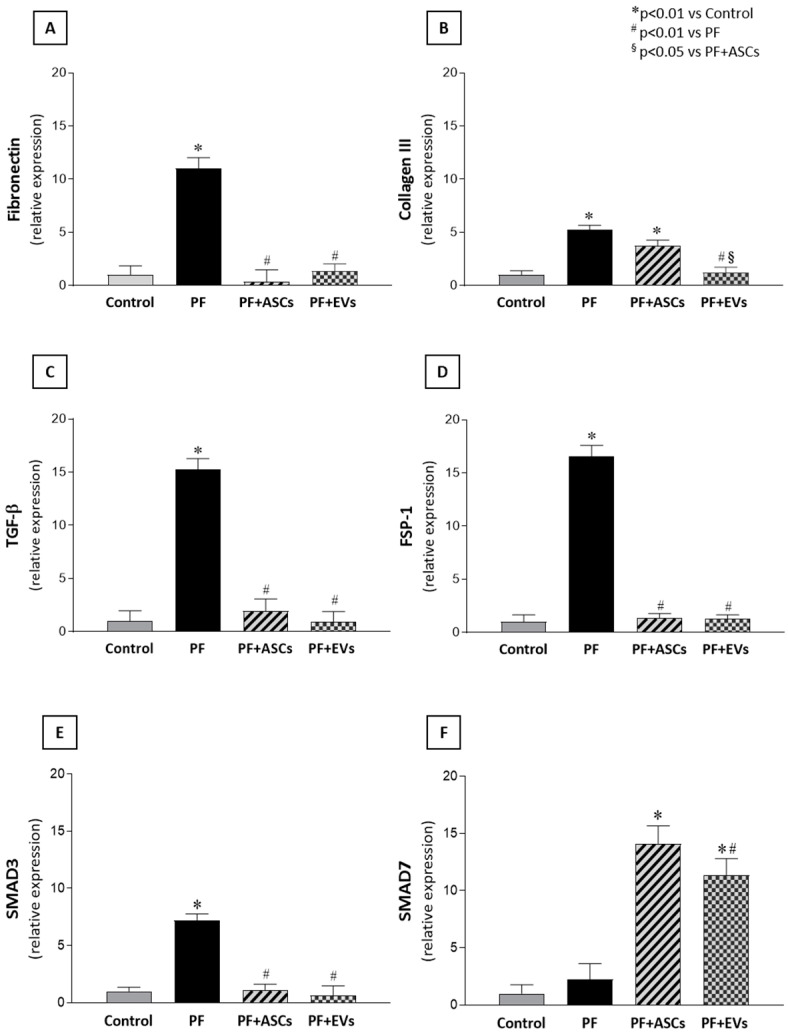
Bar graphs illustrating the comparative analysis of (**A**) fibronectin, (**B**) collagen III, (**C**) TGF-β, (**D**) FSP-1, (**E**) Smad3, and (**F**) Smad7 mRNA levels measured by quantitative real-time polymerase chain reaction in PM samples of the studied groups: control (*n* = 6), PF (*n* = 8), PF+ASCs (*n* = 7), and PF+EVs (*n* = 8). A one-way ANOVA statistical analysis was performed: * *p* < 0.01 vs. control, # *p* < 0.01 vs. PF, § *p* < 0.05 vs. PF+ASCs.

**Figure 4 cells-14-00436-f004:**
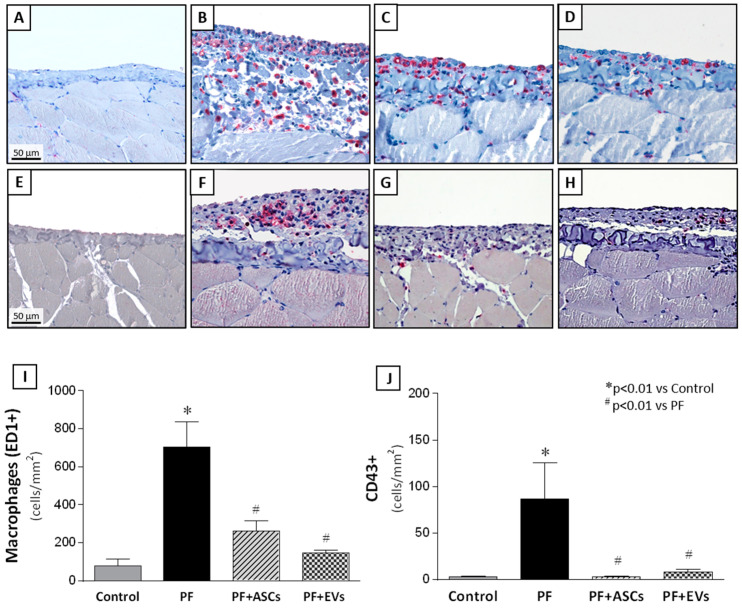
Representative photomicrographs of immunohistochemistry for macrophages (CD68 clone ED-1) and leucocytes (CD43) in PM samples of the studied groups (×400). (**A**) Macrophages in the PM in the control rats (*n* = 6) were rare. (**B**) In the PF samples (*n* = 8), significant macrophage infiltration was detected and prevented by treatment with ASCs (*n* = 7) (**C**) or EVs (*n* = 8) (**D**). Similar results were observed in relation to leukocyte infiltration in the PM. (**E**) Only a few leukocytes were present in normal PMs, whereas a marked number of leukocytes was identified in the thickened PMs (**F**). Treatment with ASCs (**G**) or EVs (**H**) abrogated the inflammatory cell infiltration. Quantitative analysis of (**I**) macrophage and (**J**) leucocyte infiltration in the peritoneal area of the animals in the experimental groups are presented as bar graphs. A one-way ANOVA statistical analysis was performed: * *p* < 0.01 vs. control, # *p* < 0.01 vs. PF.

**Figure 5 cells-14-00436-f005:**
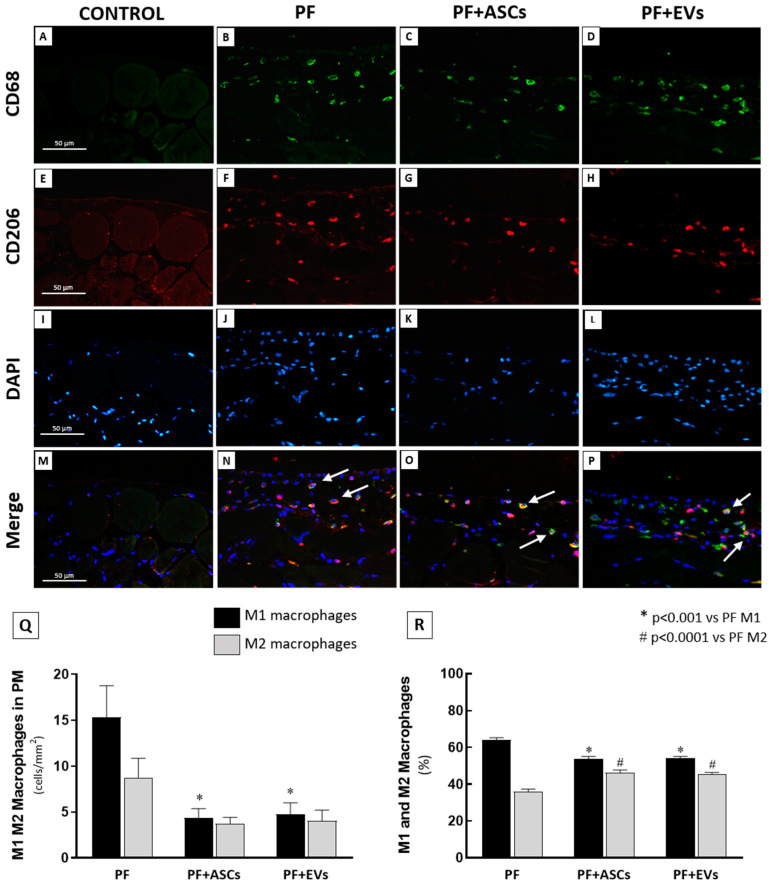
Immunofluorescence double staining for CD68 and CD206 was used to evaluate M1 and M2 macrophage infiltration in the PM. The M1 subpopulation was identified by CD68+/CD206− (green cells) and the M2 subpopulation by CD68+/CD206+ (double stained, colorful cells). (**A**–**D**) Representative photomicrographs of total CD68+ macrophages, green fluorescence. (**E**–**H**) Representative photomicrographs of CD206+ cells, highlighted in red. (**I**–**L**) Representative photomicrographs of DAPI counterstaining in blue. (**M**–**P**) Representative photomicrographs of CD68 and CD206 co-localization (white arrows), recognizing the M2 macrophage subpopulation. (**Q**) Bar graph representation of the total number of M1 and M2 macrophages (cells/mm^2^) in the PM samples of rats from the following experimental groups: PF (*n* = 5), PF+ASCs (*n* = 5), and PF+EVs (*n* = 5). (**R**) Graphic representation of the percentage of M1 and M2 macrophages in the PM of the same groups. A one-way ANOVA statistical analysis was performed: * *p* < 0.01 vs. PF M1, # *p* < 0.01 vs. PF M2.

**Figure 6 cells-14-00436-f006:**
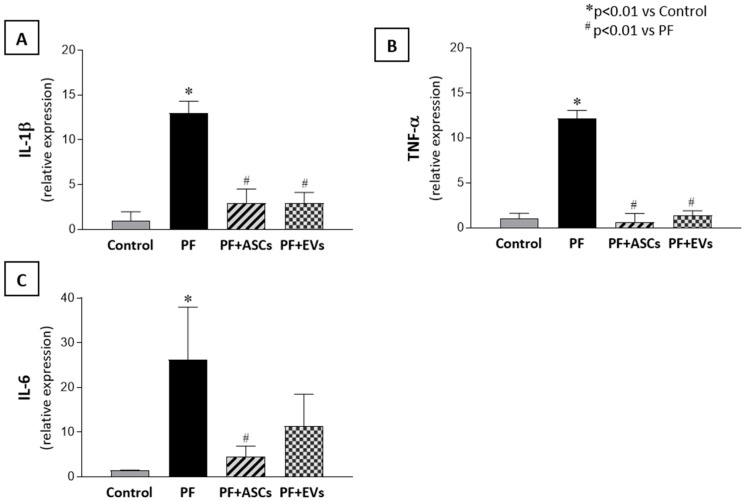
mRNA levels of pro-inflammatory mediators obtained through real-time PCR. The 18S gene was used as an internal control. (**A**) IL-1β, (**B**) TNF-α, and (**C**) IL-6 in the PM samples from the studied groups: control (*n* = 6), PF (*n* = 8), PF+ASCs (*n* = 7), and PF+EVs (*n* = 8). A one-way ANOVA statistical analysis was performed: * *p* < 0.01 vs. control, # *p* < 0.01 vs. PF.

**Figure 7 cells-14-00436-f007:**
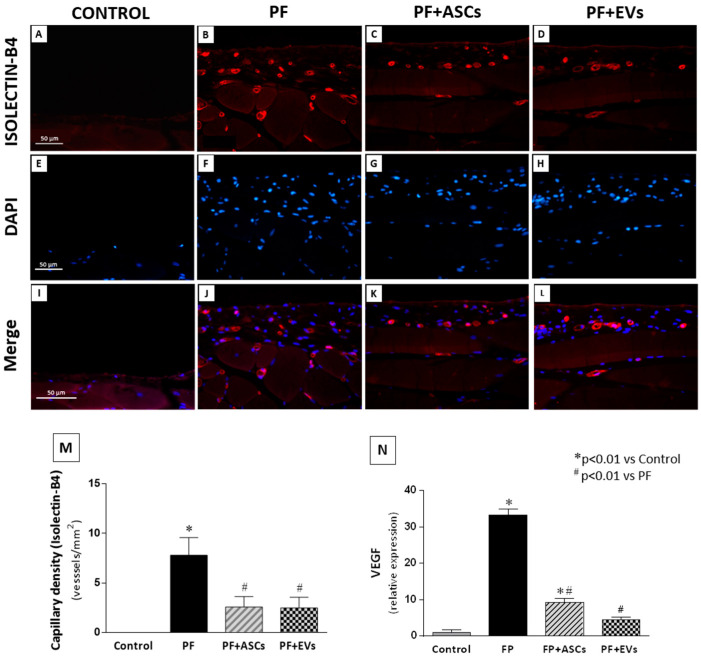
Immunofluorescence for isolectin B4 (stained in red) was used to identify the presence of neovessels, while DAPI (stained in blue) was used in the PM of the studied groups (×400). (**A**,**E**,**I**): Few blood vessels were detected in the PM of the control group (*n* = 5). (**B**,**F**,**J**) Untreated PF animals (*n* = 5) presented a marked increase in the number of neovessels in the PM. (**C**,**G**,**K**) The PF+ASCs (*n* = 5) and (**D**,**H**,**L**) PF+EVs (*n* = 5) groups showed a similar result, with less prominent capillary density compared with the untreated PF group. (**M**) Bar graph showing the vascular density across groups. (**N**) Quantitative analysis of VEGF gene expression in PM. A one-way ANOVA statistical analysis was performed: * *p* < 0.01 vs. control, # *p* < 0.01 vs. PF.

**Figure 8 cells-14-00436-f008:**
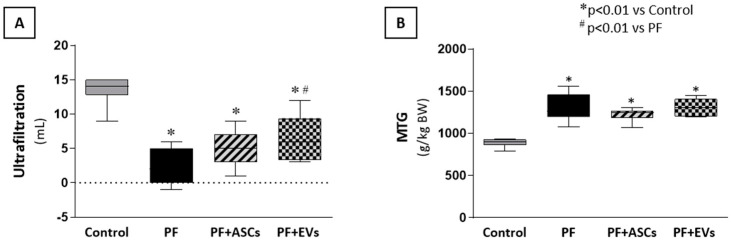
Effects of ASCs and EVs on peritoneal function evaluated through (**A**) the ultrafiltration rate (UF) and (**B**) the mass transfer of glucose (MTG) in the animals in each experimental group: control (*n* = 5), PF (*n* = 5), PF+ASCs (*n* = 5), and PF+EVs (*n* = 5). A one-way ANOVA statistical analysis was performed: * *p* < 0.01 vs. control, # *p* < 0.01 vs. PF.

## Data Availability

All data generated in the present study are included in this published article. Further methodology details and rough data tables are fully available on request from the corresponding author.

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
