# Peer review of "Adipose Tissue Stem Cells (ASCs) and ASC-Derived Extracellular Vesicles Prevent the Development of Experimental Peritoneal Fibrosis"

_cells, 2025, doi:10.3390/cells14060436_

Round 1
Reviewer 1 Report
Comments and Suggestions for Authors
In the research article titled “Adipose Tissue Stem Cells (ASC) and ASC-derived Extracellular Vesicles Prevent the Development of Experimental Peritoneal Fibrosis,” Gouveia et al. investigate the therapeutic potential of ASCs and ASC-derived extracellular vesicles (EVs) in a rat model of peritoneal fibrosis (PF) induced by chlorhexidine gluconate (CG). The study demonstrates that both ASCs and their derived EVs exhibit significant anti-inflammatory and anti-fibrotic effects. While the experimental design is robust and the article is well-written, it requires major revisions to further enhance clarity and address critical aspects of the research.
Major Comments:
1. The authors presented the effects of ASC and ASC-derived EVs at the end of 30 days following injury induction, concluding, as stated: “…our study has demonstrated the potential protective effects.” However, it is essential for the authors to illustrate the progression of the model over time. By doing so, they can accurately determine whether the observed effects are protective or curative. To support this, I recommend including Masson’s trichrome staining images at various time points throughout the model's development, such as T0, T3, T10, and T30. This would provide a clearer visual representation of the fibrosis progression and the treatment's impact over time. I also suggest including mRNA analysis of fibronectin, collagen III, IL-1β, and TNFα. The Figure can either be included in the main text or presented as a Supplementary Figure; I leave this decision to the authors' discretion.
2. The authors stated that the therapeutic effect of EVs was achieved using 30 μg of protein. However, the dosage of vesicles should be expressed in terms of particle number, based on nanoparticle tracking analysis (NTA) data. Please revise the manuscript accordingly.
3. A representative Western blot showing the expression of Tetraspanins CD63, CD81, and CD9 in both ASC and ASC-derived EVs is needed as part of the characterization.
4. The figure legends need to be improved. Please include the number of rats per group, the original magnification, scale bars, definitions of abbreviations and symbols, and the statistical methods used.
5. The Discussion section needs significant improvement. The authors do not address the effects of either treatment on neo-angiogenesis, nor do they explore the potential mechanisms of interaction between ASCs or ASC-derived EVs that could explain the observed outcomes. The Discussion should be rewritten to include these aspects, and it should also be revised to incorporate findings from the new figure requested.
Minor Comments:
1. Figures 5 and 7 need to be replaced with higher-quality images.
2. Several typos need correction: in Figure 3, "collagen" is misspelled, and in Figure 7, "Isolectin-B4" is incomplete, with "B4" missing.
Author Response
Reviewer 1
1) The authors presented the effects of ASC and ASC-derived EVs at the end of 30 days following injury induction, concluding, as stated: “…our study has demonstrated the potential protective effects.” However, it is essential for the authors to illustrate the progression of the model over time. By doing so, they can accurately determine whether the observed effects are protective or curative. To support this, I recommend including Masson’s trichrome staining images at various time points throughout the model's development, such as T0, T3, T10, and T30. This would provide a clearer visual representation of the fibrosis progression and the treatment's impact over time. I also suggest including mRNA analysis of fibronectin, collagen III, IL-1β, and TNFα. The Figure can either be included in the main text or presented as a Supplementary Figure; I leave this decision to the authors' discretion.
Answer: We thank the referee for these insightful suggestions. Detailed responses to both parts of the comment are as follows:
1. Progression of the Model Over Time:
We agree that illustrating the progression of fibrosis at intermediate time points (e.g., T3 and T10) could provide valuable insights into the temporal dynamics of the treatment's effects. However, our experimental design focused on evaluating treatment outcomes at the endpoint of 30 days. Animals in the T0 (control group), representing the baseline conditions before injury induction, were sacrificed at the same time as those in the other groups, ensuring that all groups were analyzed under consistent and comparable conditions.
Introducing intermediate time points would require replicating all experimental groups to ensure comparability and maintain consistency across experimental conditions. Without such replication, comparisons could lead to misleading results due to variability in experimental conditions. Replicating all groups at additional time points would significantly increase the use of animals, resources, and time. Given the ethical principle of reducing animal use and our focus on predefined study objectives, we believe the current design adequately addresses the study’s goals. To acknowledge this limitation, we have revised the Discussion section to clarify the scope of our design and present the observed effects as protective based on the defined endpoints.
2. mRNA Analysis of Fibronectin, Collagen III, IL-1β, and TNF-α:
Answer: We performed qPCR analysis of cDNA from peritoneal samples of the studied animals to measure the relative gene expression of fibronectin, collagen III, TGF-β, FSP-1, Smad3, and Smad7, as presented in Figure 3. In addition, the relative expression of IL-1β, IL-6, and TNF-α is presented in Figure 6 and of VEGF in Figure 7N. These results were included in the first version of the manuscript. We believe these analyses fulfill the referee’s request and provide a comprehensive assessment of gene expression relevant to the experimental model.
2) The authors stated that the therapeutic effect of EVs was achieved using 30 μg of protein. However, the dosage of vesicles should be expressed in terms of particle number, based on nanoparticle tracking analysis (NTA) data. Please revise the manuscript accordingly.
Answer: Following the recommendation of the reviewer, we repeated the analysis and characterization of EVs, as now described in the 2nd paragraph of Section 2.4 on characterization of EVs. The number and size distribution of EVs were analyzed using the NANOSIGHT 3 NTA device (NanoSight Ltd). After correction for the dilution factor, we found that each EV inoculum comprised approximately 4x1011 particles, varying in size between 50 and 600 μm; particles with an approximate diameter of 250 μm were predominant (Supplementary Figure S2B(. In addition, EVs were morphologically characterized by electron transmission microscopy (electron transmission microscope model JEM 1011; JEOL, Massachusetts, USA) at 80kV and a new figure has been added in Supplementary Figure S2A.
3) A representative Western blot showing the expression of Tetraspanins CD63, CD81, and CD9 in both ASC and ASC-derived EVs is needed as part of the characterization.
Answer: We agree with the referee. In order to meet the proposed deadline for reviewing the article and avoid delay to the analysis of the new version of the manuscript, EVs were characterized by their positivity for tetraspanins CD9 and CD63 by dot blot, instead of western blot analysis. As now described in the last paragraph of Section 2.4. on characterization of EVs, 10 µL of EV solution, containing approximately 10 µg of total protein, was spotted on nitrocellulose membranes, incubated with the primary antibodies (anti-CD9 [Sigma-Aldrich #C9993] and rabbit anti-CD63 [Abcam #Ab134045]), followed by incubation with an anti-rabbit HRP secondary antibody and development with ECL chemiluminescent substrate. A new Supplementary Figure S3 shows that EV extracts were positive for both CD9 and CD63 biomarkers.
4) The figure legends need to be improved. Please include the number of rats per group, the original magnification, scale bars, definitions of abbreviations and symbols, and the statistical methods used.
Answer: The figure legends have been corrected accordingly. We have added the number of animals used per experimental group for each parameter studied and details of the statistical methods used in the analysis.
5) The Discussion section needs significant improvement. The authors do not address the effects of either treatment on neo-angiogenesis, nor do they explore the potential mechanisms of interaction between ASCs or ASC-derived EVs that could explain the observed outcomes. The Discussion should be rewritten to include these aspects, and it should also be revised to incorporate findings from the new figure requested.
Answer: We thank the referee for these helpful comments. We agree with the points raised and have revised the Discussion accordingly. The effects of both ASC and ASC-derived EVs on neo-angiogenesis have been addressed, and potential mechanisms of interaction between ASCs and EVs have been explored.
Minor Comments:
1. Figures 5 and 7 need to be replaced with higher-quality images.
Answer: We have reviewed the figures and enhanced the quality and saved them TIFF format before inserting in the manuscript template. If preferred, we can also send the open format pictures directly to the reviewer or the journal editor.
2. Several typos need correction: in Figure 3, "collagen" is misspelled, and in Figure 7, "Isolectin-B4" is incomplete, with "B4" missing.
Answer: Thank you for the careful revision. We have corrected the typos and the figure legends accordingly.
Reviewer 2 Report
Comments and Suggestions for Authors
Gouveia et al reported that adipose tissue-derived stem cells (ASCs) and extracellular vesicles (EVs) blocked the progression of peritoneal fibrosis(PF)with anti-inflammatory effects and enhanced ultrafiltration capacity in a rat model of PF induced by chlorhexidine gluconate. Followings are my comments that would be helpful to improve the manuscript.
Major:
1 As the authors described, there were several similar studies (references 9-12) before their study. What are the differences between the previous studies and this work? What is the most important finding and novelty in this work? In the Introduction section, the authors did not address this question in detail.
2 . Materials and Methods.
Line 200-202. Although the authors described the methods of α-SMA stained area (%), more information needs to be provided for this method. In addition, other quantification methods for macrophage, leucocyte infiltration, percentage of M1 macrophages and M2 macrophages on the PM, the vascular density must be described in detail.
3. Results.
Figure 5Q, the legend was “Graphic representation of macrophages on the PM double positive to CD68 and CD206”, while the figure 5Q was “M1 M2 macrophages in PF (cells/mm2). Figure 5R showed that ASCs and EVs treatment increased the percentage of M2 cells. However, line 496-497 expressed “treatment with ASCs and EV reduced both macrophage subpopulations”. This is paradoxical. To confirm it,Figure 6 should added the expression of M2 macrophages markers.
Line 303 the subsection title of the results was not consistent with the text. Line 296-311 these results should be integrated together.
4 The discussion is more of a repetitive description of the results. More should be discussed about the most important findings, but not all the results.
Minor
There were several spelling errors,such as 2 × 106, CO2.
Comments on the Quality of English LanguageNative English speakers are advised to revise grammar and spelling issues
Author Response
Reviewer 2
1) As the authors described, there were several similar studies (references 9-12) before their study. What are the differences between the previous studies and this work? What is the most important finding and novelty in this work? In the Introduction section, the authors did not address this question in detail.
Answer: Thank you for this important observation. We have revised the relevant paragraph in the Introduction to better highlight the differences. Studies 9–11 used experimental models of peritoneal fibrosis (PF), and study 12 was a phase 1 clinical trial focusing on the safety of ASCs, but none of these studies compared the effects of extracellular vesicles (EVs) derived from ASCs with the cells themselves. Our work addresses this gap by exploring the therapeutic potential of EVs versus ASCs in an experimental PF model.
The revised paragraph now reads:
“Stem cell-based therapies also offer an alternative approach to quell the inflammatory and fibrotic processes, potentially inducing regeneration of the PM [9–11]. Although previous studies demonstrated the feasibility and therapeutic effects of adipose tissue-derived stem cells (ASCs) in experimental models [9–11] and clinical applications targeting PF [12], the potential of extracellular vesicles (EVs) derived from ASCs compared with the cells themselves remains insufficiently explored. EVs, as a cell-free therapeutic strategy, retain the paracrine regenerative effects of ASCs and offer advantages such as improved scalability for therapeutic production and enhanced stability for storage and transport, which are critical for future clinical applications.”
2) Materials and Methods: Line 200-202. Although the authors described the methods of α-SMA stained area (%), more information needs to be provided for this method. In addition, other quantification methods for macrophage, leucocyte infiltration, percentage of M1 macrophages and M2 macrophages on the PM, the vascular density must be described in detail.
Answer: We would like to thank the referee for the careful revision of our text. We have now revised the Materials and Methods section according to these comments.
Immunohistochemistry quantification
First paragraph of Section 2.8. on immunohistochemistry and immunofluorescence: The percentage area of PM occupied by α-SMA was calculated by measuring the total PM area and the positive PM area (stained in red after immunohistochemistry development) of at least 20 microscopic fields from each studied animal using Image-Pro Plus 7.0 software (Media Cybernetics). We used these 2 measurements to calculate the % α-SMA for each microscopic field and used the arithmetic mean of all fields from the same animal to obtain the individual % α-SMA. Peritoneal macrophages and the inflammatory infiltrate population evaluated by immunohistochemistry were counted in at least 20 microscopic fields/animal, under 200× magnification, and expressed as cells/mm2.
Immunofluorescence quantification
Second paragraph of Section 2.8 on immunohistochemistry and immunofluorescence: To evaluate the presence of M1 and M2 macrophage subtypes in the PM samples from the experimental groups, 20 epifluorescence microscopic fields were analyzed per animal, under 200× magnification, using the NIS-Elements Software (Nikon). Each microscopic field was captured under 3 different light filters (blue, green, and red), thus producing 3 microphotographs for each field. Using the "merge" tool of the NIS-Elements software, a 4th picture per field was produced with an overlay of all colors. The M1 macrophage subpopulation was recognized as CD68+/CD206− cells, visible in the merged pictures as green cells. The number of green cells was counted and expressed as M1 cells/mm2 (black bars on the graph in Figure 5Q). M2 macrophages were recognized as CD68+/CD206+ cells, visible in the merged pictures as colorful yellowish cells due to the overlay of green, red, and blue from the nuclei in the same cell. The number of colorful cells was counted and expressed as M2 cells/mm2 (gray bars on the graph in Figure 5Q). To express the percentage of M1 and M2 macrophages in PM samples, we used the following calculations: (M1 x 100)/total macrophages and (M2 x 100)/total macrophages, respectively.
Section 2.9 on quantitative assessment of angiogenesis in the revised version of our manuscript: The endothelial marker, DyLight 594-labeled Griffonia simplicifolia isolectin B4 (Vector Laboratories), was used to detect new blood vessels in PM samples by immunofluorescence. For this purpose, 10 microscopic fields were scored under 400× magnification using a 594-nm filter for isolectin and a 540-nm filter for DAPI (nuclei). The number of isolectin-B4-positive blood vessels was counted and the density of capillaries present in each slide was determined by the number of blood vessels divided by the area of PM and expressed as the number of vessels/mm2.
3) Results: Figure 5Q, the legend was “Graphic representation of macrophages on the PM double positive to CD68 and CD206”, while the figure 5Q was “M1 M2 macrophages in PF (cells/mm2). Figure 5R showed that ASCs and EVs treatment increased the percentage of M2 cells. However, line 496-497 expressed “treatment with ASCs and EV reduced both macrophage subpopulations”. This is paradoxical. To confirm it,Figure 6 should added the expression of M2 macrophages markers. Line 303 the subsection title of the results was not consistent with the text. Line 296-311 these results should be integrated together.
Answer: There was a mistake in our previous Figure 5R. We would like to thank the reviewer again for the careful revision, because it allowed us to see the error and replace the bar graphs with the correct ones. We also corrected the description of these results in the text accordingly. As now detailed in the Results section, although reducing the global macrophage infiltration (represented by the sum of M1 and M2 macrophage subtypes) in the PM samples of PF rats (Figure 5Q), only the reduction of the M1 subpopulation was statistically different in PF+ASCs or PF+EVs compared with untreated PF. The reduction in M2 subpopulation promoted by ASCs or EVs was numerical only, suggesting relative conservation of M2 macrophages in the treated animals. Accordingly, the bar graphs in Figure 5R show that the percentage of M2 macrophages in untreated PF animals (36.1%) seems to have been increased by the treatments with ASCs and EVs (46.3% and 45.7%, respectively).
4.The discussion is more of a repetitive description of the results. More should be discussed about the most important findings, but not all the results.
Answer: We thank the referee for the valuable feedback. We have revised the Discussion to focus more on the most important findings, rather than reiterating the results. The revised version highlights the key insights from the study and provides a more in-depth analysis of the main outcomes.
Minor Comments:
- There were several spelling errors,such as 2 × 106, CO2.
- Native English speakers are advised to revise grammar and spelling issues.
Answer: Thank you for the suggestion. The manuscript has been reviewed and revised by a native English speaker to ensure proper grammar, spelling, and clarity.
- Submit a graphical abstract that summarizes your manuscript:
= A Graphical Abstract is provided and has been uploaded in the submission platform. A copy of our Graphical Abstract is shown below (in the attached version of "Answers to the reviewers)
Round 2
Reviewer 2 Report
Comments and Suggestions for Authors
After a major revision, the quality of the revised manuscript has been improved greatly. The authors has answered the questions well. The manuscript is suggested to be accepted.